# One-Pot Synthesis of Epirubicin-Capped Silver Nanoparticles and Their Anticancer Activity against Hep G2 Cells

**DOI:** 10.3390/pharmaceutics11030123

**Published:** 2019-03-15

**Authors:** Jun Ding, Guilin Chen, Guofang Chen, Mingquan Guo

**Affiliations:** 1Key Laboratory of Plant Germplasm Enhancement & Specialty Agriculture, Wuhan Botanical Garden, Chinese Academy of Sciences, Wuhan 430074, China; sophia_dj@126.com (J.D.); glchen@wbgcas.cn (G.C.); 2Key Laboratory of Analytical Chemistry for Biology and Medicine (Ministry of Education), Department of Chemistry, Wuhan University, Wuhan 430072, China; 3Sino-Africa Joint Research Center, Chinese Academy of Sciences, Wuhan 430074, China; 4Chemistry Department, St. John’s University, Queens, New York, NY 11439, USA

**Keywords:** epirubicin, one-pot synthesis, silver nanoparticle, antitumor

## Abstract

Epirubicin-capped silver nanoparticles (NPs) were synthesized through a one-pot method by using epirubicin as both the functional drug and the reducing agent of Ag^+^ to Ag^0^. The preparation process was accomplished in 1 h. In addition, the obtained epirubicin-capped silver nanoparticle was characterized by transmission electron microscopy (TEM), energy-dispersive X-ray spectroscopy (EDX), and infrared spectroscopy. The results showed that a layer of polymer epirubicin had formed around the silver nanoparticle, which was 30-40 nm in diameter. We further investigated the antitumor activity of the prepared epirubicin-capped silver nanoparticle, and the half maximal inhibitory concentration (IC_50_) against Hep G2 cells was 1.92 μg/mL, indicating a good antitumor property of the nanoparticle at low dosage.

## 1. Introduction

Nanoparticles (NPs), ranging from 1 to 100 nm in size, have gained considerable interest because of their various potential applications in clinical nanomedicine and research [1,2,3]. Noble metal NPs, especially silver NPs, with the properties of large specific surface area, enhanced surface plasmon resonance, and minimal side effects have been proposed for potential use in the anticancer area [4,5]. For instance, Ag NPs of approximately 50 nm in size can target tumor tissues based on the enhanced permeation and retention (EPR) effect [6,7], and the photo-thermal conversion in Ag NPs can increase the temperature locally, thereby achieving therapeutic killing of tumor cells [8,9,10]. Clearly, Ag NPs represent a promising anticancer agent with the assistance of photo-thermal therapy. However, the therapeutic effect of bare Ag NPs is limited. The functionalization of Ag NPs with anticancer drugs is a promising method to provide synergistic effects by combining both therapeutic actions, resulting in better anticancer effects.

Typically, drug-loaded Ag NPs are fabricated by top-down and bottom-up approaches. In the top-down method, a silver NP core is first obtained, and then post-modification and drug-loading steps are performed [11,12]. Obviously, the process is complicated and time-consuming, and the morphology control of the final particles is also troublesome. As for bottom-up methods, drugs could be incorporated directly into Ag NPs during the growth of starting materials, which is more convenient. However, the chemical reducing agent required by the method, such as NaBH_4_, pyridine, etc., might lead to the presence of toxic reagents existing on or in Ag NPs [13], which would bring about adverse effects in further biological or medical applications. Moreover, the recently reported bottom-up methods mostly chose to load plant extract onto Ag NPs [14,15,16], possibly resulting in an undefined curative effect or side effect. Therefore, a new green and simple strategy to prepare drug-capped Ag NPs is highly favorable.

Epirubicin (Figure 1) is an anticancer prescription medicine for chemotherapy [17], which is proved to be effective to treat breast, lung, and liver cancers. It possesses several phenolic hydroxyl groups, thus exhibiting a weak reduction property. In this study, we employed epirubicin as the functional drug and also the reducing agent, thereby developing a novel, green, and mild method to synthesize epirubicin-capped silver NPs by the chemical reaction process between the silver salt solution and epirubicin without using any surfactant. Subsequently, the antitumor effect of the resultant NPs was evaluated.

## 2. Materials and Methods

### 2.1. Chemicals and Reagents

Silver nitrate (99.8%) was obtained from Aladdin (Shanghai, China). Epirubicin (98%) was purchased from Melonepharma (Dalian, China). Fetal calf serum (FCS) was purchased from Sijiqing Co. (Hangzhou, China). Dulbecco’s modified eagle medium was supplied by Gibco (Waltham, MA, USA). 3-(4,5-Dimethylthiazol-2-yl)-2,5-diphenyltetrazolium bromide (MTT) was obtained from Biosharp (Hefei, China). Ultra-pure water used throughout the study was purified with the Milli-Q system (Millipore, Bedford, MA, USA).

### 2.2. Preparation of Epirubicin-Capped Silver NPs

Epirubicin-capped silver NPs were obtained using a green and mild procedure [18,19]. First, a AgNO_3_-MilliQ water solution (10^−2^ M) was heated till boiling at 120 °C under magnetic stirring. Subsequently, an epirubicin solution was added to obtain the final concentration of 10^−4^ (*w*/*v*). After reacting at 120 °C for 1 h, the resultant epirubicin-capped silver NPs were obtained. Subsequently, excess unreacted epirubicin was removed by dialysis in ultra-pure water for 48 h [20,21]. The final product was obtained by centrifugation at 10,000× *g* for 30 min.

### 2.3. Characterization of Epirubicin-Capped Silver NPs

TEM images were obtained using JEM-100CXII transmission electron 135 microscopes (TEM, Jeol, Tokyo, Japan). FT-IR spectra were obtained using an AVATAR 137 360 FT-IR spectrometer (Thermo Electron Inc., San Jose, CA, USA). The chemical composition of epirubicin-capped silver NPs was examined by EDX-720 energy-dispersive X-ray analysis (EDX, Shimadzu, Kyoto, Japan) using Mg Kα radiation as the excitation source.

### 2.4. Cytotoxicity Assay

The cytotoxic effect of the epirubicin silver NPs at various concentrations against Hep G2 cells was evaluated by the MTT assay [22], and the absorbance in MTT assay was measured by a BioTek EL-x800 microplate reader (Bio-Tek Instruments Inc, Winooski, VT, USA). In brief, approximately 30,000 Hep G2 cells were seeded in a 96-well flat-bottom tissue culture plate. After cultivation for 24 h, they were exposed to the NPs at concentrations ranging from 0 to 30 µg/mL, and epirubicin at concentrations in the range of 0–50 µg/mL, respectively for 48 h. After treatment, the supernatant was removed and washed with phosphate-buffered saline (PBS), and then a 20 µL portion of MTT reagent (5 mg/mL) was added to each well and incubated overnight at 37 °C for 4 h to permit the formation of purple formazan crystals. Subsequently, the supernatant was discarded, and a 150 µL portion of DMSO was added in the well to dissolve the purple formazan crystals. Finally, the absorbance was measured at 570 nm using a microplate reader. The inhibition rate was calculated by subtracting the absorbance of samples treated with silver NPs or epirubicin from that of the negative controls, and then comparing it to the absorbance of the negative controls.

## 3. Results and Discussion

### 3.1. Characterization

The morphology of the as-synthesized NPs was investigated by transmission electron microscopy (TEM). The TEM image (Figure 2) shows that the NPs exhibit spherical and uniform morphologies with an average diameter of 36 mm, and a layer of film is observed to cap around the NPs, which was believed to be a layer of poly-epirubicin and epirubicin complex formed during the Ag^+^ reduction process. The resultant Ag NPs are of a suitable size to be retained among the tumor tissues, which makes them a good passive targeting agent.

To further demonstrate the successful preparation of epirubicin-capped silver NPs, the composition was examined by energy-dispersive X-ray spectroscopy (EDX). As shown in Figure 3, the NPs were clearly constituted by Ag, C, and O. The existence of C and O indicates organic compounds were incorporated into the obtained NPs, which is ascribed to poly-epirubicin or epirubicin.

Furthermore, infrared spectroscopy was employed to understand the formation mechanism of the epirubicin-capped silver NP. The IR spectra of epirubicin and epirubicin-capped silver NPs were obtained, respectively. Figure 4A shows the IR spectrum of epirubicin. The presence of two peaks at 2920 cm^−1^ and 2850 cm^−1^ is assigned to the stretching vibration of the CH_2_ group, whereas the presence of the absorption peak at 1385 cm^−1^ belongs to the bending vibration of CH_2_. The peak at 1632 cm^−1^ is attributed to a contemporaneous contribution of C=C and C=O stretching mode, and the broad peaks at 3435 cm^−1^ correspond to the O–H stretching vibration of water. Specifically, the signal at 1276 cm^−1^ and 1210 cm^−1^ is derived from the ν (C–O) of the enol function. From the IR spectrum of epirubicin-capped silver NPs as shown in Figure 4B, the existing peaks in the range of 3000–1300 cm^−1^ were similar to that of epirubicin, indicating the successful coating of epirubicin onto silver NPs. In particular, the signals at 1276 cm^−1^ and 1210 cm^−1^, attributable to the C–O stretching mode of the enol, disappeared, which might be ascribed to the oxidation of the phenolic hydroxyl groups into quinone or phenolic ether. It is believed that during the oxidation process of epirubicin, epirubicin acted as a reducing reagent, which converted Ag^+^ into silver NPs, and in the meantime epirubicin was capped onto the surface of the silver NPs, leading to the successful formation of the resultant epirubicin-capped Ag NPs. It is also worth mentioning that after standing the prepared epirubicin-capped silver NPs in water for 30 days at room temperature, there was no change in morphology or composition observed (data not shown), indicating a good stability of the prepared nanoparticles.

### 3.2. Evaluation of Antitumor Activity

To demonstrate the antitumor activity of the obtained epirubicin-capped silver NPs, the cytotoxicity of the Ag NPs was investigated using the 3-(4,5-dimethylthiazol-2-yl)-2,5- diphenyltetrazolium bromide (MTT) method, and Hep G2 cells were chosen as the target cell line. To this end, Hep G2 cells were cultivated with free epirubicin and epirubicin-capped silver NPs in a series of diverse concentrations, respectively. After cultivation for 72 h, the cell viabilities were obtained. Based on this result, the concentrations of IC_50_ for both free epirubicin and epirubicin-capped silver NPs toward Hep G2 cells were calculated from the survival curves. As shown in Figure 5 and Table 1, the IC_50_ values of epirubicin-capped silver NPs and epirubicin were 1.92 μg/mL (hill slope at 2.43) and 0.11 μg/mL (hill slope at 0.57), respectively [23,24,25], which means that the Ag NPs gave a remarkably increased IC_50_ value compared with free epirubicin, but still fell well within the proper range of most anticancer drug candidates in terms of IC_50_ values. The obtained hill slopes also indicated that epirubicin-capped NPs exhibited positive cooperativity, whereas epirubicin exerted negative cooperativity of binding. The difference in binding behavior between epirubicin-capped silver NPs and free epirubicin could be ascribed to the incorporation of silver NPs with epirubicin. In fact, when applying free epirubicin directly onto the cancer patient, several side effects and toxicity were observed as a relatively large dose of the drug was taken at one time, leading to a sudden rise and rapid fall of the drug concentration in blood. In the case of using epirubicin silver NPs as the antitumor drug, the silver NPs could target the tumor tissues based on EPR because of the suitable size of the NPs, making the normal tissue free from the drug. Furthermore, epirubicin was wrapped onto the silver NPs in the form of a polymer and it could be slowly released to the tumor tissue, thus significantly lowering the cytotoxicity compared to free epirubicin [26,27]. This result convincingly demonstrates that the as-prepared Ag NPs are promising antitumor agents at low dosage. Moreover, we compared the performance of the prepared epirubicin-capped silver NPs with other epirubicin-loaded systems. According to the data summarized in Table 2, our system exhibits superior cytotoxic activity with lower IC_50_ values [28,29,30].

## 4. Conclusions

In summary, we have presented a one-pot synthesis method for fabricating epirubicin-capped silver NPs. The reaction was accomplished by the fast chemical reaction between a silver salt solution and epirubicin without using any surfactant, thus making the method simple, green, and mild. The resultant epirubicin-capped silver NPs exhibited a great anticancer property at low dosage. Considering the mechanisms of the synthesis method, the universality of the proposed method should be further evaluated, which is an ongoing study in our laboratory. Taken together, this approach represents a new option for preparing drug-loaded Ag NPs with a broad application prospect in clinic.

## Figures and Tables

**Figure 1 pharmaceutics-11-00123-f001:**
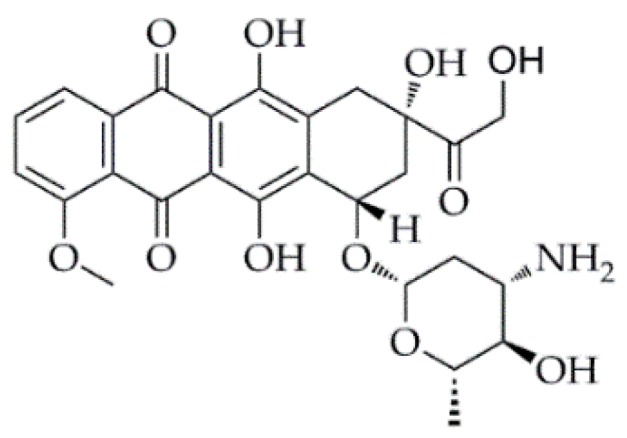
Structure of epirubicin.

**Figure 2 pharmaceutics-11-00123-f002:**
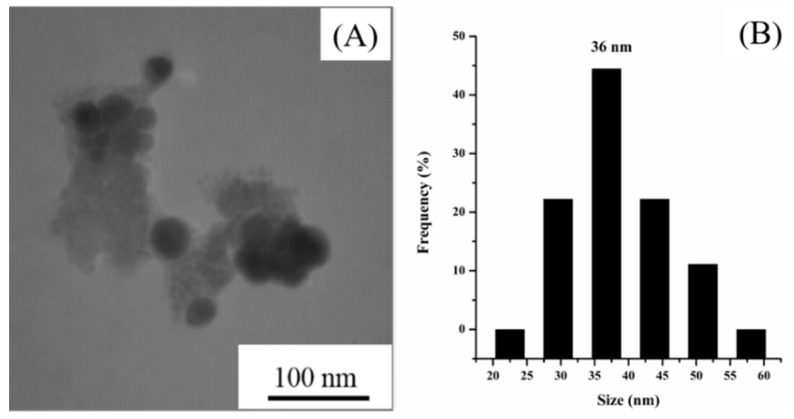
(**A**) Image and (**B**) size distribution diagram of epirubicin-capped silver nanoparticles (NPs).

**Figure 3 pharmaceutics-11-00123-f003:**
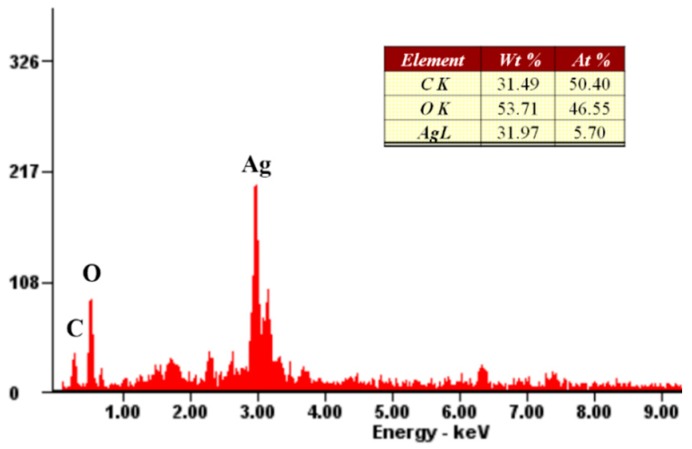
Energy-dispersive X-ray spectroscopy (EDX) spectrum and chemical composition (the inset table) of epirubicin-capped silver NPs.

**Figure 4 pharmaceutics-11-00123-f004:**
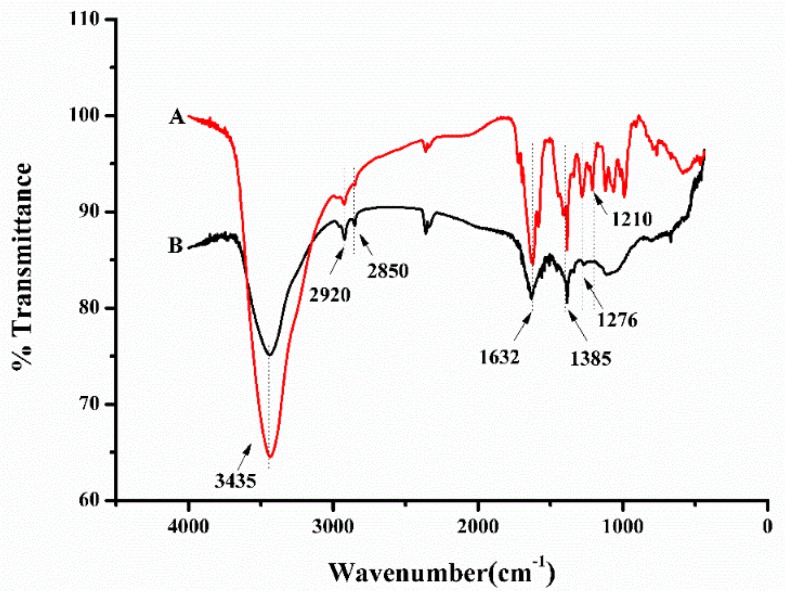
FT-IR spectra of (**A**) epirubicin and (**B**) epirubicin-capped silver NPs.

**Figure 5 pharmaceutics-11-00123-f005:**
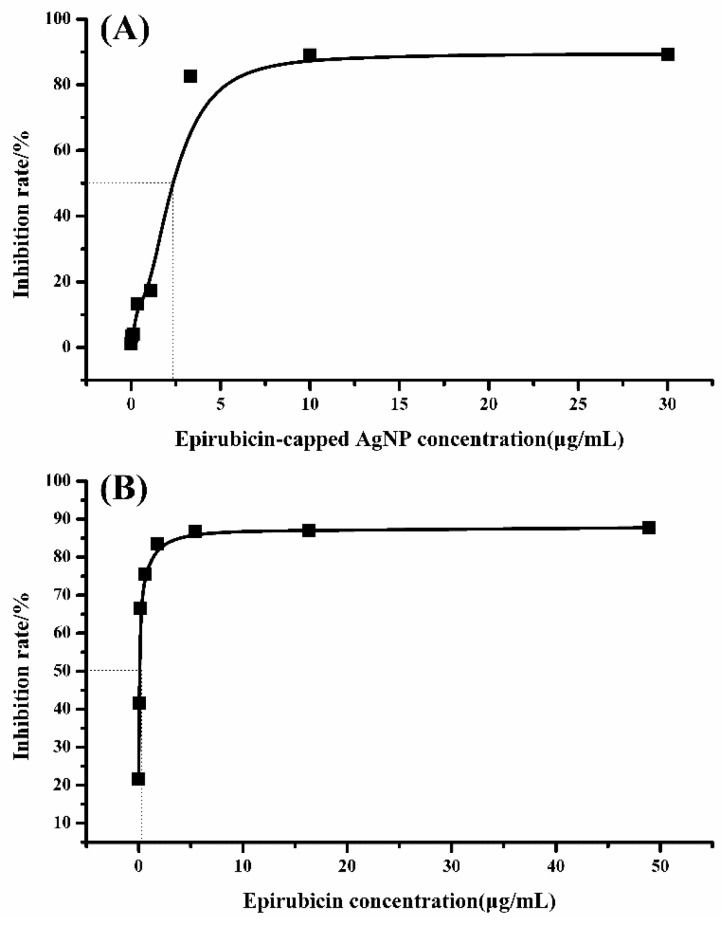
Inhibition curve of (**A**) epirubicin-capped silver NPs and (**B**) epirubicin for the Hep G2 cell line.

**Table 1 pharmaceutics-11-00123-t001:** IC_50_ and hill slopes of epirubicin-capped silver NPs and epirubicin.

Drug	IC50 (μg/mL)	Hill Slope
Epirubicin-capped silver NPs	1.92	2.43
Epirubicin	0.11	0.57

**Table 2 pharmaceutics-11-00123-t002:** Comparison of cytotoxic activities of reported epirubicin drug systems.

Nanoparticle	Cell	IC_50_	Reference
Epirubicin-loaded folate-modified PA nanoparticles	Human nasopharyngeal epidermal carcinoma cell	2.75 μg/mL	[28]
Epirubicin-loaded PLGA/TPGS nanoparticles	Hep G2 cells	0.78 μg/mL	[29]
Epirubicin-loaded poly(lactic-*co*-glycolic acid) nanoparticles	MCF-7 cells	824 nM	[30]
Epirubicin-capped silver NPs	Hep G2 cells	0.11 μg/mL	This work

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
