# Peer review of "One-Pot Synthesis of Epirubicin-Capped Silver Nanoparticles and Their Anticancer Activity against Hep G2 Cells"

_pharmaceutics, 2019, doi:10.3390/pharmaceutics11030123_

Reviewer 1 Report

The paper represents the green synthesis of epirubicin capped AgNPs and study of their cytotoxicity. The paper should be improved before accepted for publication. In my opinion, the obtained results represent the novel method for obtaining epirubicin capped AgNPs with antitumor properties, but the experimental design suffers from the lack of some methods needed for proper characterization of obtained AgNPs.

 May main remarks are as follows:

The structural formula of epirubicin should be included.

The additional spectrophotometric measurements should be performed in order to better characterise the process of AgNPs synthesis. The absorption spectra of epirubicin capped and bare AgNPs should be presented. The best ratio of epirubicin/AgNPs concentration in the experiments which lead to the composite formation should be evaluated. Some data of stability of epirubicin capped AgNPs are missing.

In TEM characterization of morphology and size of AgNPs, the distribution size dependent diagram should be included. 

 Besides, the inhibition curves should be described by appropriate mathematical models (Hill or some other equation ) and the obtained parameters of inhibition curves should be presented in the appropriate Table. The literature supported conclusion and discussion about the lower inhibition power, i.e. cytotoxicity of epirubicin capped vs.bare AgNPs must be included.

In general, the discussion about the obtained results including their comparison with the similar model systems is missing.

 The additional remarks: Sections 2.2 and 2.4, the references concerning dialysis and cytotoxicity assay should be added. Section 2.3 - the last sentence should be moved to section 2.4.

Author Response

The structural formula of epirubicin should be included.

Response: Thanks for the comment. The structure of epirubicin was provided in Figure 1 of the revised manuscript.

The additional spectrophotometric measurements should be performed in order to better characterise the process of AgNPs synthesis. The absorption spectra of epirubicin capped and bare AgNPs should be presented. The best ratio of epirubicin/AgNPs concentration in the experiments which lead to the composite formation should be evaluated. Some data of stability of epirubicin capped AgNPs are missing.

Response: Thanks for the advice. Actually, we’ve tried to use the spectrophotometric measurement to guide the synthetic process of epirubicin capped silver NPs. However, the UV spectra of the epirubicin solution and the epirubicin capped silver NP solution were almost identical, as they both showed strong UV absorbance around 400 nm. That made the monitoring of the synthetic process using spectrophotometric measurement unsuitable for our case. The ratio of drug/AgNPs concentration was selected according to previous papers with minor modification (Materials Letters 92 (2013) 350-353, Nanoscale, 2014,6, 10113-10117). And the description of the stability of the particles was described in Line 125-127 in the revised manuscript.

In TEM characterization of morphology and size of AgNPs, the distribution size dependent diagram should be included.

Response: Thanks for the suggestion. The distribution size dependent diagram of the prepared nanoparticles was illustrated in Figure 2B of the revised manuscript.  

Besides, the inhibition curves should be described by appropriate mathematical models (Hill or some other equation) and the obtained parameters of inhibition curves should be presented in the appropriate Table. The literature supported conclusion and discussion about the lower inhibition power, i.e. cytotoxicity of epirubicin capped vs.bare AgNPs must be included.

Response: Thanks for the advice. The inhibition curves were plotted using Hill equation, and the parameters were summarized in Table 1 of the revised manuscript. Also, the literature reporting the cytotoxicity of epirubicin, epirubicin loaded NPs, and silver NPs were included as ref 23-24.

In general, the discussion about the obtained results including their comparison with the similar model systems is missing.

Response: Thanks for the advice. We compared the performance of the prepared epirubicin capped silver NPs with other epirubicin loaded systems. According to the data summarized in Table 2 of the revised manuscript, our system exhibits superior cytotoxic activity with lower IC50 values.

The additional remarks: Sections 2.2 and 2.4, the references concerning dialysis and cytotoxicity assay should be added. Section 2.3 - the last sentence should be moved to section 2.4.

Response: Thanks for the suggestion. As the reviewer requested, the references were added as ref 20-21, and the last sentence of Section 2.3 was moved to Section 2.4. 

Reviewer 2 Report

The authors. One-pot synthesis of epirubicin capped silver 2 nanoparticles and their anticancer activity against 3 Hep G2 cells: characterization, Cytotoxicity assay: is a very interesting work. However, I recommend this manuscript to be published in Pharmaceutics with a few recommendations.

Line 56 the authors wrote "Taxol" but after that the taxol do not appear any more.

Line 76.  Figure 1. "After cultivation for24 h" should be " After cultivation for 24 h "

Line 106. The peak at 3400 was not assigned

The authors do not show any data about the stability of the particles 

Author Response

Comments and Suggestions for Authors

The authors. One-pot synthesis of epirubicin capped silver 2 nanoparticles and their anticancer activity against 3 Hep G2 cells: characterization, Cytotoxicity assay: is a very interesting work. However, I recommend this manuscript to be published in Pharmaceutics with a few recommendations.

Line 56 the authors wrote "Taxol" but after that the taxol do not appear any more.

Response: Sorry for the mistake. The sentence related to “taxol” was deleted in the revised manuscript.

Line 76.  Figure 1. "After cultivation for24 h" should be " After cultivation for 24 h "

Response: Thanks for pointing out the typo. It was corrected in the revised manuscript.

Line 106. The peak at 3400 was not assigned

Response: Thanks for the comment. The peak at 3400 was assigned to the O-H stretching vibration of water, and the peak was also labeled in revised figure 4.  

The authors do not show any data about the stability of the particles 

Response: Thanks for the advice. The stability of the particles was described in Line 125-127 in the revised manuscript. 

Reviewer 3 Report

The manuscript “One-pot synthesis of epirubicin capped silver nanoparticles and their anticancer activity against Hep G2 cells” explains chemotherapy drug-coated AgNPs. This seems to be an important study and deserves the publication in Pharmaceutics with minor comments as above;

1.       Structure of the drug must be given.

2.       Suitable graphical abstract can be encouraged.

3.       Authors should explain the purification step done for epirubicin capped silver NP before the treatment to cancer cell line.

4.       Discuss in detail the anti-cancer mechanism by epirubicin capped silver NP.

Author Response

Comments and Suggestions for Authors

The manuscript “One-pot synthesis of epirubicin capped silver nanoparticles and their anticancer activity against Hep G2 cells” explains chemotherapy drug-coated AgNPs. This seems to be an important study and deserves the publication in Pharmaceutics with minor comments as above;

 1. Structure of the drug must be given.

Response: Thanks for the advice. The structure of epirubicin was provided in Figure 1 of the revised manuscript.

2. Suitable graphical abstract can be encouraged.

Response: Thanks for the advice. A graphical abstract was added now.

3. Authors should explain the purification step done for epirubicin capped silver NP before the treatment to cancer cell line.

Response: Thanks for the advice. After the epirubicin capped silver NP was synthesized, the excess free epirubicin was removed by dialysis in ultra-pure water for 48 h. Please see in Line 68-70 for the revision.

4. Discuss in detail the anti-cancer mechanism by epirubicin capped silver NP.

Response: Thanks. According to the suggestion by the reviewer, the anti-cancer mechanism of epirubicin capped silver NP was discussed. Please see in Line 142-150 of the revised manuscript.

Round  2

Reviewer 1 Report

Please explain briefly in Section 3.1. what is the difference between the two TEM mages in Fig.2 A. The size distribution in Fig.2 B should be better represented as a histogram?

English must be improved. I am not native in English, but for example, I am convenient that the first word in the sentence cannot be "And"... Please take care that the rating of the selected journal is quite high.

In figure caption (Fig.4), a and b should be replaced by capital letters. 

Please, indicate which data support the conclusion regarding NPs stability and the morphology at the end of Section 3.1.

Please add the reference concerning Hill equation, or include this equation in the text. 

Please add the explanation of n coefficient values. n describes the cooperativity of the process, it seems that epirubicin capped NPs exerted high cooperativity and epirubicin exert negative cooperativity in interaction, i.e. after binding one molecule of ligand to the binding site, the affinity of binding of other ligand decreased.  

Author Response

Comments and Suggestions for Authors

Please explain briefly in Section 3.1. what is the difference between the two TEM mages in Fig.2 A. The size distribution in Fig.2 B should be better represented as a histogram?

Response: Thanks. The two TEM images were both that of epirubicin capped silver NPs. For better understanding, one of them was removed. And the size distribution in Fig. 2B was changed to a histogram in the revised manuscript.

English must be improved. I am not native in English, but for example, I am convenient that the first word in the sentence cannot be "And"... Please take care that the rating of the selected journal is quite high.

Response: Thanks for pointing out the issue. English was improved in the revised manuscript.

In figure caption (Fig.4), a and b should be replaced by capital letters.

Response: Thanks. The mistake has been corrected in the revised manuscript.

Please, indicate which data support the conclusion regarding NPs stability and the morphology at the end of Section 3.1.

Response: Thanks. The morphology was observed by TEM after obtaining the NPs for 30 days, and no change was found. Therefore, we think it’s unnecessary to provide new TEM images. And we noted “data not shown” in the revised manuscript.

Please add the reference concerning Hill equation, or include this equation in the text.

Response: Thanks for the suggestion. References were added in the revised manuscript.

Please add the explanation of n coefficient values. n describes the cooperativity of the process, it seems that epirubicin capped NPs exerted high cooperativity and epirubicin exert negative cooperativity in interaction, i.e. after binding one molecule of ligand to the binding site, the affinity of binding of other ligand decreased. 

Response: Thanks for the suggestion. The explanation of n coefficient values were discussed in Line 139-142 of the revised manuscript.